# Prediction of Machine Failure in Industry 4.0: A Hybrid CNN-LSTM Framework

**Abdul Wahid** [1,*][ID], **John G. Breslin** [1][ID] and **Muhammad Ali Intizar** [2][ID]

1   Data Science Institute (DSI), National University of Ireland Galway, H91 TK33 Galway, Ireland; john.breslin@nuigalway.ie
2   School of Electronic Engineering, Dublin City University, D09 W6Y4 Dublin, Ireland; ali.intizar@dcu.ie
*   Correspondence: a.wahid2@nuigalway.ie

**Abstract:** The proliferation of sensing technologies such as sensors has resulted in vast amounts of time-series data being produced by machines in industrial plants and factories. There is much information available that can be used to predict machine breakdown and degradation in a given factory. The downtime of industrial equipment accounts for heavy losses in revenue that can be reduced by making accurate failure predictions using the sensor data. Internet of Things (IoT) technologies have made it possible to collect sensor data in real time. We found that hybrid modelling can result in efficient predictions as they are capable of capturing the abstract features which facilitate better predictions. In addition, developing effective optimization strategy is difficult because of the complex nature of different sensor data in real time scenarios. This work proposes a method for multivariate time-series forecasting for predictive maintenance (PdM) based on a combination of convolutional neural networks and long short term memory with skip connection (CNN-LSTM). We experiment with CNN, LSTM, and CNN-LSTM forecasting models one by one for the prediction of machine failures. The data used in this experiment are from Microsoft's case study. The dataset provides information about the failure history, maintenance history, error conditions, and machine features and telemetry, which consists of information such as voltage, pressure, vibration, and rotation sensor values recorded between 2015 and 2016. The proposed hybrid CNN-LSTM framework is a two-stage end-to-end model in which the LSTM is leveraged to analyze the relationships among different time-series data variables through its memory function, and 1-D CNNs are responsible for effective extraction of high-level features from the data. Our method learns the long-term patterns of the time series by extracting the short-term dependency patterns of different time-series variables. In our evaluation, CNN-LSTM provided the most reliable and highest prediction accuracy.

**Keywords:** production forecasting; Industry 4.0; manufacturing production line; convolutional neural network (CNN); artificial intelligence (AI); long short-term memory (LSTM); 1D Convolution; skip-connection; smart manufacturing; Internet of Things (IoT); predictive maintenance (PdM)

## 1. Introduction

In industry, maintenance is of crucial importance, as it directly impacts the cost, reliability, ability, quality, and performance of a company. Unwanted or unplanned downtime of equipment adds to the degradation and suspension of the core business, resulting in immeasurable losses and significant penalties.n It was reported that Amazon suffered a set back of $4 million in sales for a downtime of 49 min in 2013 [1]. According to a survey by the Ponemon Institute [2], on average an organization loses $138,000 per hour because of downtime. Also, it is reported that the operations and maintenance costs of offshore wind turbines amount to 20% to 35% of their total generated revenue [3], and for the oil and gas industry the maintenance expenditure can range from 15% to 70% of the total production cost [4]. It is therefore crucial for industries to look into efficient and competent methods to prevent unexpected downtime, hence reducing operation and maintenance costs and improving reliability.

With the evolution of technology like the Internet of Things (IoT) [5], we are able to connect manufacturing devices to networks to send and exchange data. The first step for industry is to connect their machines and gather data from them. The data generated are collected and stored in the cloud and can later be utilized for analysis and visualization. Predictive maintenance has been discussed and researched for some time [6], but in recent years it has become more widely accessible because of the seamless advancements in modern technology [7]. It generally involves monitoring the operating conditions of the machines and gathering the information that could be used to maximize the interval time between repairs and to minimize unscheduled disruption due to machine failures [8]. Modern technology has the potential to identify and detect developing faults in machine components, predict a fault's progression, and finally to provide a strategy for maintenance schedules.

Predictive maintenance is an important part of smart manufacturing and Industry 4.0. According to a recent report, the market for PdM will be worth $23 billion by 2026 [9]. The purpose of PdM is to identify uncommon machine behavior and to send out a warning about probable system damage. However, it still remains one of the key challenges faced by the industry. Additionally, with the advent of various technologies, many new concepts have been reinforced for PdM such as using IoT to gather increasing amounts of data from machines equipped with sensors [10], advanced techniques for data pre-processing, e.g., feature extraction, normalization, data cleaning, and preparation, etc., and machine learning and deep learning-based models for condition monitoring and failure detection. All of these improvements and discoveries have improved overall system reliability, reducing machine downtime and costs. Currently, most modern machines deployed in manufacturing production lines are equipped with advanced sensors and increased communications capabilities. These machines are capable of monitoring their status by continuously observing the conditions of various important features such as voltage, pressure, vibration, rotation, temperature, motor speed, etc. These features of a modern machine are used to train deep learning algorithms that are capable of detecting and predicting faults even before they occur. This allows us to augment or even replace an ordinary/regular maintenance schedule through predictive maintenance, which is more reliable and advanced, reduces unnecessary maintenance and its associated costs, and guarantees the proper and timely functioning of machines.

Deep learning models have provided effective solutions in fault diagnosis due to their powerful feature learning abilities. They build a set of representation methods using numerous layers and learn the non-linear representation of any time series to a higher level of complexity and abstraction. Many papers have been published that cover machine fault diagnosis to a large extent over the past few years. A detailed overview of a deep learning-based monitoring system for machine health was proposed in [11]. A simple auto-encoder, CNN and RNN-based machine health monitoring system was proposed in [12]. A prognostic and health management architecture based on deep learning was proposed in [13]. An auto encoder-based CNN for induction motor diagnosis was also proposed by [14]. A gated CNN to estimate the remaining life of a machine was proposed in [15]. Wu et al. [16] used LSTM to predict the remaining life of turbofan engines.

Deep learning has been studied to predict an unknown time series using historical data [17]. A variety of modern machine learning algorithms ranging from multi-layer perceptrons (MLP) and long short-term memory (LSTM) to the combinations of RNNs and dynamic Boltzmann machines [17] have been used in time-series forecasting. All of these algorithms have their own strengths and weaknesses, depending on the type of data or the task at hand. However, due to the complex nature of predictive maintenance use cases, none of the above noted algorithms have provided excellent results. Moreover, hybrid methods [18] are a new development trend in the field of time-series forecasting.

In this paper, we propose a hybrid deep learning framework based on CNN-LSTM for PdM. The proposed method combines the advantages of convolutional neural networks (CNN), which extract the effective features and patterns among multi-variate input

variables, and long short term memory (LSTM), which captures the complex long-term dependencies and automatically selects the mode best suited for relevant time-series data. The hybrid CNN-LSTM model is designed to make the optimization easier by extracting the effective features using CNNs and then incorporating LSTM in parallel to predict machine failures. The main contributions of our work are summarised as follows:

- We propose a hybrid deep learning model (CNN-LSTM) for PdM. The model uses CNN to extract features from the time series and LSTM for prediction. It uses a time sequence of different errors and analyses the correlation between different input variables for better prediction.
- We introduce a novel temporal skip connection component for LSTM that enables them to capture long length dependencies and makes optimization easier and efficient.
- On comparing the evaluation indexes of CNN, LSTM, and hybrid CNN-LSTM, we found that our hybrid CNN-LSTM has the highest prediction accuracy and is more reliable and suitable for PdM forecasting.

The IoT sensors and CCTVs are responsible for collecting data from the different available equipment on a production floor. Efficient use of data with AI encourages innovation, better productivity, and technology dissemination.

Structure of the Paper: The rest of the paper is organized as follows. Section 2 provides related work of the state-of-the-art data-driven approaches. Section 3 throws some light on the Industry 4.0 use case. Sections 4 and 5 provides some background to existing methodologies, including CNN and LSTM, used in the design of the model and presents our framework by explaining the structural design of the model, respectively. Section 6 explains the results and the optimization strategy adopted in this study. Section 7 explains the lesson learned throughout the course of this study. Section 8 discusses the limitations of the proposed framework, and, finally, Section 9 concludes the paper.

## 2. Related Work

Data-driven approaches have attracted many research efforts in the area of smart manufacturing. Machine learning techniques for manufacturing applications, along with their weaknesses and strengths, are explained in [19,20]. Successful machine learning algorithms such as Bayesian Networks, artificial neural networks, and other ensemble methods are explained in [21]. Traditional machine learning algorithms such as logistic regression, artificial neural networks, or support vector machines yield a modest performance because of their shallow structural design and hand-crafted feature engineering [22]. Deep learning has exhibited impressive performance in fields such as image classification, natural language processing, semantic segmentation, and object detection/recognition [23]. Deep learning is best known for automatically extracting highly nonlinear and complex abstract features by means of multiple layers stacked on top of each other [24]. Because of its automated feature extraction and ability to learn different levels of data abstractions, deep learning can identify hidden patterns and trends in data and predict them through a well-defined optimization pipeline. The authors of [25] introduced long short-term memory (LSTM), a special kind of RNN, capable of dealing with complex data and learning long-term dependencies. Over time, these networks were further optimized in other work [26,27]. LSTMs have demonstrated their success in time-series forecasting [28]. Recently in [29], a CNN combined with K-means, which was proposed for time-series load forecasting, achieved impressive results. The main reason for this is the capability of a CNN to extract features from input data at different levels. Because a CNN has the ability to learn for a vast range of non-linear problems as explained in [22], it is well suited for time-series production data. In contrast, LSTMs are effective in modelling time-series data because of their ability to remember the states of input data in their memory [28]. Deep learning method with LSTM combined with support vector machine (SVM) proposed in [30] are utilized to distinguish aberrant data from normal vibration signals gathered during a reduction gearbox endurance test and helicopter test flight data acquired by many sensors located throughout the aircraft. A method based on JDA and deep belief network (DBN) for

fault bearing diagnosis was proposed in [31]. The JDA is used in the JACADN to perform feature transfer between source domain samples and target domain samples, i.e., the kernel function maps the source domain samples and target domain samples into the same feature space. Anomaly detection using a generative adversarial network (GANS) for industrial sensor data was proposed in [32]. The paper focused on reconstruction of sound signal reconstructions and detection of anomalies. The basic notion underlying the techniques is that normal conditions can be accurately reconstructed using a smaller latent space interim of neural network design, whereas abnormal conditions cannot be reproduced due to larger reconstruction losses. This method is appropriate for anomaly detection, as the volume of anomalous condition data is typically much smaller than normal condition data, and a detection model may be trained solely using normal condition data.

The use of deep learning techniques to detect anomalies has improved the results of older methods. An artificial neural network model underpins deep learning. Deep learning and LSTMs offer to train hierarchical models over input data that describe probability distributions. Artificial intelligence has become a vibrant subject with many practical applications such as machine prognosis, fault detection, predictive maintenance, etc., thanks to recent advancements in both hardware and brain models, particularly in the previous decade.

## 3. Industry 4.0 Use Case

Consider a typical use case of a smart factory having its production lines equipped with modern sensing technologies. All data produced by the sensors installed on production lines are acquired and accessible in real time. As depicted in Figure 1, a variety of modern AI techniques can be applied to the real-time data collected from sensors to get better insights into factory operations in real time.

In a given smart manufacturing production line, the objective is to have an active response to demand and supply. Deep learning technologies can help with maximizing the efficiency of real-time processes when analyzing large amounts of data. They can be used to quickly analyse the relevant data and identify failures that can result in reduced production over a subsequent period of time. They provide information about the status of the equipment involved, thus supporting the human workforce maintaining and monitoring this equipment [33]. They also provide oversight for machine conditions and hence guarantee more reliability and process regularity, which eventually diminishes the necessity for regular and constant machine checkups and also reduces machine downtime [34,35].

In a typical smart production line, data from a large number of devices such as sensors and CCTVs (as shown in Figure 1) are collected and then processed into a more human-understandable form. The collected data are stored in scalable data storage systems. Big data techniques are used to pre-process and analyze the data. The data analysis can be presented through rich visualizations. Once the data are prepared, deep learning algorithms such as LSTMs are used, which identify the correlations between different features and take the appropriate steps based on the data. LSTMs are applied to the sensor data or time-series data to find anomalies within a production line. The LSTMs predict critical errors in advance to avoid a shutdown, fatal accident, or any unwanted event. The basic difference between LSTMs and other existing models is in terms of being more proactive and reactive. LSTMs are responsible for predictive maintenance, which can help many smart production lines to save millions of dollars in the support and maintenance of equipment. CNNs perform automatic feature extraction and can learn a high order representation of data (e.g., time-series or image-based data) [13]. However, CNNs alone when applied to time-series data are highly reliant on data pre-processing and tuning of a large pool of hyperparameters. In contrast, CNNs combined with LSTMs identify the most important features that contribute to the output. The combined model does not require heavy feature engineering, which makes the model computationally more reliable and efficient when compared to standalone CNNs.

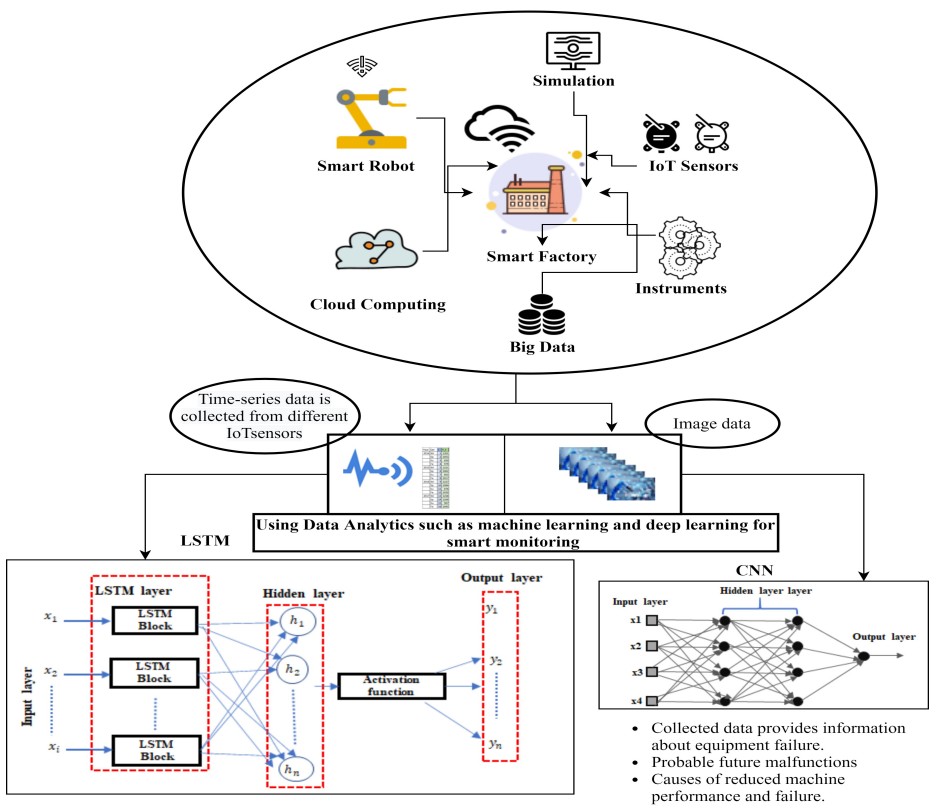

**Figure 1.** A connected smart manufacturing production line for Industry 4.0. The data collected from IoT sensors are efficiently used with AI innovations, better productivity, and technology dissemination. Deep learning algorithms such as LSTMs are used for time-series forecasting, and CNNs are used for processing image-based data for predictive maintenance. The overall goal of the framework is to have more information processed over the Internet of Things.

## 4. Framework

This section presents an overview of the proposed hybrid CNN-LSTM framework. We have designed the framework specifically for multivariate time-series classification. In the section below we explain the building blocks of our hybrid model in detail. We first formulate the PdM problem, and then discuss in detail the proposed hybrid CNN-LSTM architecture, and finally explain the optimization approach. The overall outline of the predictive maintenance framework is shown in Figure 2.

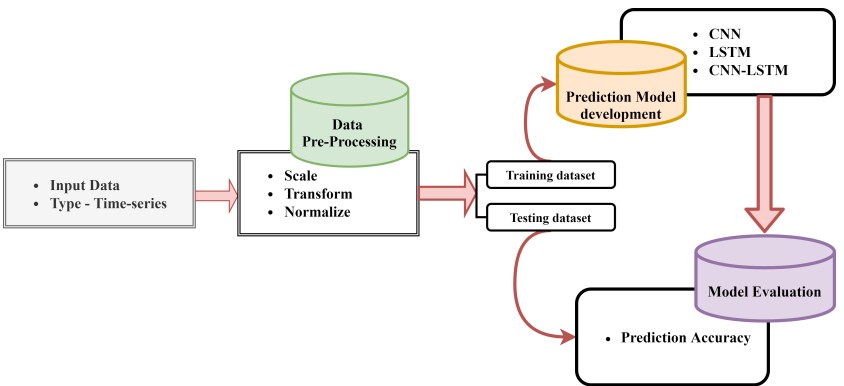

**Figure 2.** The overall architecture of the proposed predictive maintenance framework.

### 4.1. Problem Formulation

Consider a task for a multivariate time series, where when a certain time series is given, the task is to compute when machine conditions are at a state of required repair or even component replacement so that maintenance can be performed exactly when and how it is more effective. A time series is split into input data $X$ and output data $Y$. Input data are in the form of $X = [x_1, x_2, x_3, \ldots, x_t]$, where $x_t \in [0, N]$, and $N$ is the number of measurements taken in a time sequence. Our goal is to predict future repairs in a rolling forecasting fashion. To predict $y_{t+h}$, where $t$ is the current timestamp and $h$ is the future timestamp for which a prediction is to be made, we need an input sequence as $X = [x_1, x_2, \ldots, x_t]$. Similarly to predict $y_{t+h+1}$, we need the input sequence as $X = [x_1, x_2, x_t, \ldots, x_{t+1}]$. Thus the input matrix at time $t$ is prepared as $X_t = [x_1, x_2, \ldots, x_t]$.

### 4.2. Convolutional Neural Networks (CNN)

Over the past few years, CNNs have achieved remarkable results in various areas including image classification, speech recognition, natural language processing, and time-series forecasting. They were first proposed for image processing [36]. They work on the principle of general matrix multiplication in at least in one of its layers [37]. The first layer of our hybrid model is the CNN layer, and the goal is to extract the short-term patterns and dependencies between different input variables. The convolutional layer consists of multiple convolutional filters denoted by $t$ of width $x$ and height $l$. The $t$-th filter scans through the input matrix $X$ and produces

$$k_t = tanh(a_t * X + b_t) \tag{1}$$

where $k_t$ is the output function, $X$ is the input vector, and $b_t$ is the bias. The output matrix is denoted by $q_c$. The output is then fed into the LSTM layer described in Section 4.3.

### 4.3. Long Short-Term Memory (LSTM)

Because recurrent neural networks (RNNs) were known to suffer from the vanishing gradient problem, the long short-term memory (LSTM) was an improvement over them as explained in [25]. The improvement was the introduction of a gating function into the state dynamics of RNNs. LSTMs use a hidden vector $h$ and a memory vector $m$ that store long-term information. The memory vector allows interaction with the outputs from the previous state and the next input state to select which input vector should be updated or maintained. All this is achieved by means of three gates: a forget gate $f_t$, an input gate $i_t$, and an output gate $o_t$, as shown in Figure 3. The output of the convolutional layer and the input at the current timestamp are given as input to the forget gate, and the results from the forget gate are achieved using the following formula:

$$f_t = \sigma(x_t W_{xf} + h_{t-1} W_{hf} + b_r) \tag{2}$$

where $\sigma$ is the sigmoid function, $f_t$ ranges from (0, 1), $W_f$ is the weight of the forget gate, $x_t$ is the input of the current layer at time $t$, and $h_{t-1}$ is the output at the last time. The output at the last time is combined with the input at the current time and is given as input to the input gate, and as a result the output and the candidate cell output is obtained using the following formulas:

$$i_t = \sigma(W_i * [h_{t-1}, x_t] + b_i) \tag{3}$$

$$C_t = tanh(W_c * [h_{t-1}, x_t] + b_c) \tag{4}$$

where $i_t$ ranges from (0, 1), $W_i$ is the weight of input gate, $b_i$ is the input gate bias, $W_c$ is the weight of the candidate gate, and $b_c$ is the candidate gate bias. The current cell is updated in the following manner.

$$C_t = f_t * C_{t-1} + i_t * C_t \tag{5}$$

where $C_t$ ranges from $(0, 1)$. The input to the output gate at time $t$ is the output $h_{t-1}$ and input $x_t$, and the output $o_t$ is computed using the following formula:

$$o_t = \sigma(W_o[h_{t-1}, x_t] + b_o) \tag{6}$$

The output of the LSTM cell is given by calculating the output of the output gate and the cell state using the following formula:

$$h_t = o_t * tanh(C_t) \tag{7}$$

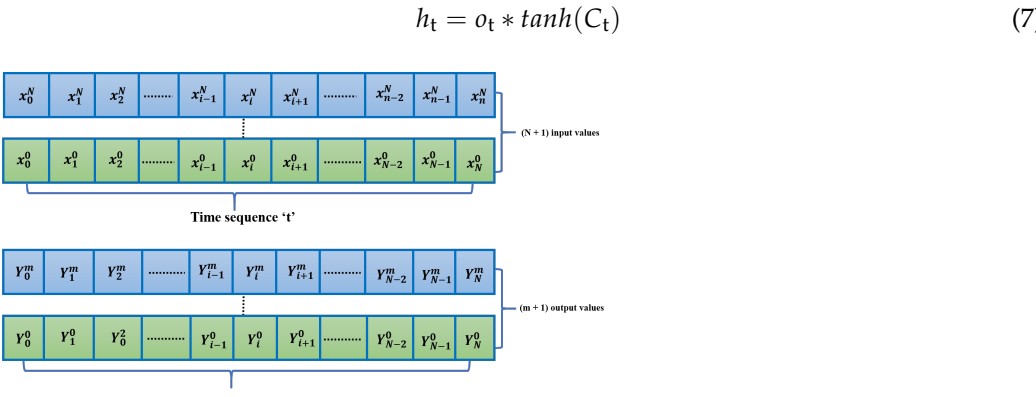

**Figure 3.** Overview of the input data $x_t$ over a time sequence $t$ with $n + 1$ values and output data $Y_t$ over a time sequence $t$ with $m + 1$ values.

## 5. Hybrid CNN-LSTM Model for PdM

In this section, we present our proposed hybrid CNN-LSTM framework, which is a two stage framework for predictive maintenance use cases. The overall structure of the proposed model is depicted in Figure 4. Our model is a combination of CNN and LSTM with skip connections (CNN-LSTM), as depicted in Figure 4. Below, we briefly summarize the dataset used and the underlying building blocks of our proposed model.

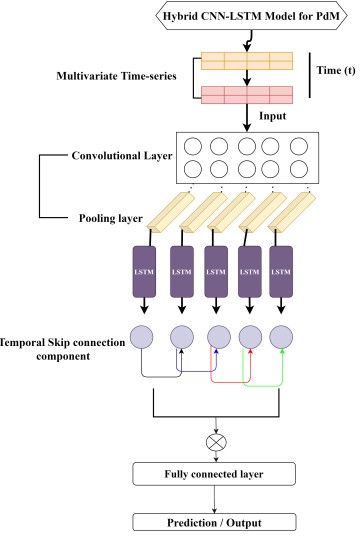

**Figure 4.** Illustration of proposed hybrid CNN-LSTM model for predictive maintenance.

### 5.1. Dataset

The data used in this experiment are from Microsoft's case study. The dataset provides information about the failure history, maintenance history, error conditions, and machine features and telemetry, which consists of information such as voltage, pressure, vibration,

and rotation sensor values. The entire dataset is summarized in Table 1. The data are collected for a total of 100 machines from 1 January 2015 to 1 January 2016. For each machine, data are collected on an hourly basis for one complete year. Therefore, for a total of 100 machines and each machine, we have (1 ∗ 365 ∗ 24) 8760 and (8760 ∗ 100) 876,000 observations, respectively. Figure 5 displays the distribution of the machine features, Figure 6 shows the failure history, and Table 2 shows the different error conditions. A total of 3919 error records are present in the dataset. A total of five error conditions are present that represent different error modes. Each error condition has an associated machineID and the timestamp as shown in Table 2. In addition, the frequency of occurrence for each errorID can be seen in Figure 7. It can be observed that the most prevalent error conditions are 1 and 2, followed by 3, 4, and 5.

The unprocessed data degrade the performance of the predictions and make learning very complex, so it is essential to standardize the data. In our experiment, we adopted Min-Max normalization. It helps us in finding different trends in the time series by comparing the features of different data points. The given time series is transformed within a definite range using normalization. It guarantees better training results during prediction by bringing the data close to zero, which makes the optimization more stable. The data values are shifted and re-scaled to range in [0, 1].

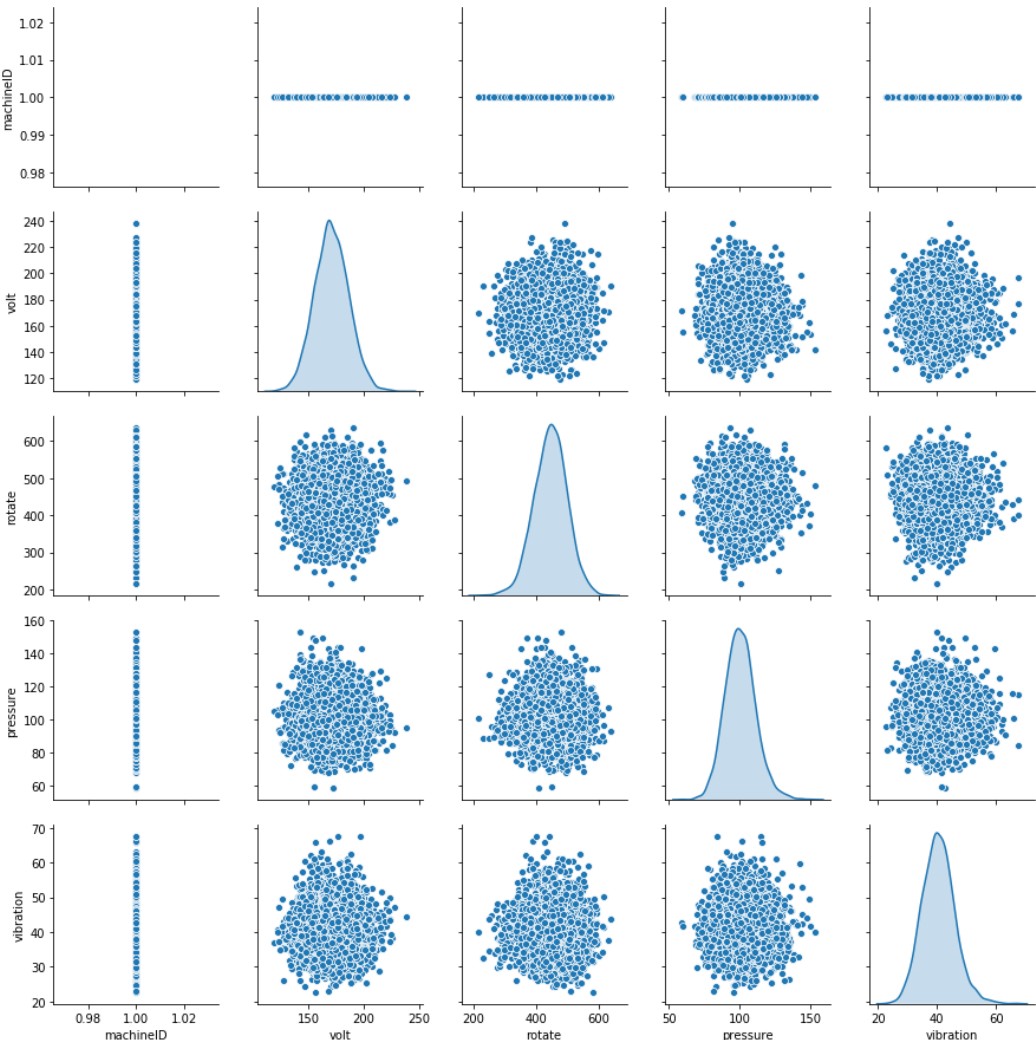

**Figure 5.** Distribution of the different machine features including machinesIDs.

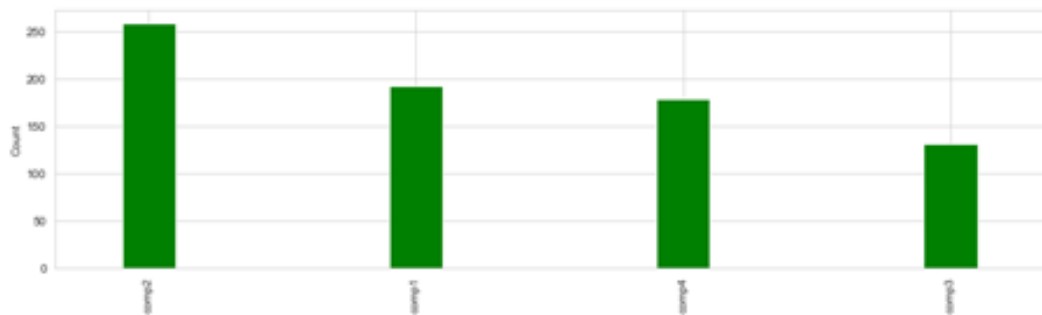

**Figure 6.** Records of component replacements that have occurred as a result of failures. A date and time stamp, machineID, and failed component type are all included in each entry.

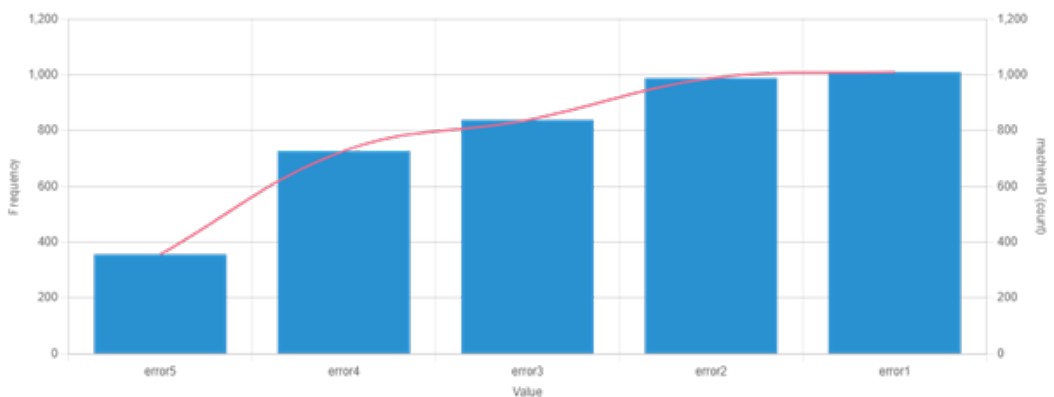

**Figure 7.** Frequency of different error conditions along with a set of 100 machines.

**Table 1.** A detailed description of different data sources.

| Data Source | Description |
| --- | --- |
| Telemetry | Time-series data of voltage, rotation, pressure, and vibration measurements recorded in real time from 100 machines and averaged across every hour throughout 2015. |
| Errors | Non-breaking errors that machines throw while in operation. |
| Maintenance | Scheduled and unscheduled records corresponding to inspection and failures of different components. |
| Machines | Includes machine summary such as model, age, and years in service. |
| Failures | Records of the replacements due to failures. |

**Table 2.** Different errorIDs with the respective machineID and datetime.

| Datetime | MachineID | ErrorID |
| --- | --- | --- |
| 2015-01-03 07:00:00 | 1 | error1 |
| 2015-01-03 20:00:00 | 1 | error3 |
| 2015-01-04 06:00:00 | 1 | error5 |
| 2015-01-10 15:00:00 | 1 | error4 |
| 2015-01-22 10:00:00 | 1 | error4 |
| 2015-01-12 14:00:00 | 2 | error4 |
| 2015-05-30 20:00:00 | 2 | error5 |
| 2015-11-20 23:00:00 | 3 | error1 |

$$X' = \frac{X - X_{\min}}{X_{\max} - X_{\min}} \tag{8}$$

where $X$ represents the value of a time series at any time instant, $X_{\min}$ and $X_{\max}$ represent the features with minimum and maximum values. According to the above equation, the

value of $X$ is scaled in the range $[0, 1]$ when the value of $X$ lies between the maximum and minimum values. It was also observed that machine breakdowns are usually rare events of the assets as compared to normal operation in predictive maintenance. This generates an imbalance in the label distribution, resulting in poor performance because algorithms prefer to classify majority class examples over minority class examples, as the total misclassification error is greatly reduced when the majority class is correctly classified. Although accuracy can be high, this results in low recall rates, which becomes a bigger issue when the cost of false alarms is considerable. To address this issue, simple strategies such as oversampling minority cases [38], were considered. The data are split: 70% training set, 10% validation set, and 20% test set, as shown in Table 3. Finally the proposed model is evaluated on the test dataset.

**Table 3.** Train, validation, and test data splits.

| Training Set | Validation | Test Set |
|:---:|:---:|:---:|
| 70% | 10% | 20% |

### 5.2. Label Generation

When utilizing multi-class classification to predict asset failure due to an issue, labeling is done by selecting a time window before the asset failure and labeling the records that fail within that window as "failure", whereas all other records are labeled as "normal". The time range is chosen based on business use case, such as in some cases predicting failure hours in advance may be sufficient, whereas in others, days or weeks may be required to allow for the arrival of new components. In our study we used a time-window of 24 h as our goal is to determine the likelihood that a machine would fail in the near future due to component breakdown. The dataset provides information of specific component failures such as component 1, 2, 3, and 4. We create a categorical feature that serves as label like failure = comp1, failure = comp1, and so on.

### 5.3. Feature Engineering

To construct features based on the properties of individual data sources, we use a lag feature engineering method [39]. The purpose of this feature creation is to identify variable inputs and dynamic lags that aid in capturing regular trends in time-series data and to create new features from the input variables that explicitly represent the time-series components. The first data source telemetry data includes timestamps, used to calculate lagging features. We choose a 3-h lag window size for the lag features to be constructed, then compute the mean, standard deviation, minimum, and maximum values to represent the time-series data's short-term history over the lag window. The extracted features are merged to create a final dataset for telemetry. Similarly, for error data source we count each type of error in a lagging window and reformat the data to have one entry for each machine with the timestamp at which the error occurred. Finally, we calculate the total number of errors of each type that occurred over a period of 24 h, at a sampling rate of 3 h. Maintenance records are an important data source, as they have the number of replacements for each component in the last three months. We calculate the duration since the last component was replaced, which gives us the information about the component failure. The days since the last component was replaced are calculated for each type of component as features from the maintenance data source. The machine features are used without any modification. All the features are merged together to create a final dataset. Finally, the labels are generated to estimate the probability of a machine failing due to a certain component. We create a categorical failure feature label for each row. The final dataset consists of 34 features for 100 individual machines collected for a period of one complete year with a sampling rate of 3 h. Therefore, for a total of 100 machines, we have 291,641 observations.

### 5.4. Hybrid CNN-LSTM

This section explains the proposed hybrid CNN-LSTM model as shown in Figure 4. The model contains a sequence of convolutional layers and a sequence of LSTM blocks that are combined using a fully connected layer at the final stage to produce an output. The input data are first passed through a convolutional layer and a pooling layer, and the corresponding features are extracted. The obtained features are given as input to the LSTM block, which is designed in such a way that it memorizes the historical data and the relative long-term dependencies. The traditional LSTM fails to capture long-term correlations; therefore we introduce a novel temporal skip connection that takes advantage of the regular pattern in input variables. For instance, the dataset used in this work shows a clear pattern on a daily basis. Based on the records at time $t$ in historical data, our model can make predictions regarding the machine failure. Regular LSTMs can barely capture this type of information because of the long length of the past time-frame (24 h) and the subsequent optimization process. Thus we introduce an LSTM with a temporal skip connection capable of prolonging the temporal span of the information flow, thereby making optimization efficient and easy. The skip connection is added between the current hidden cells and the hidden cells in the same phase in an adjacent period. The updating process is formulated as follows:

$$
\begin{aligned}
v_t &= \sigma(z_t W_{zv} + q_{t-x} W_{qv} + b_v) \\
u_t &= \sigma(z_t W_{zu} + q_{t-x} W_{qu} + b_u) \\
y_t &= tanh(z_t W_{zy} + v_t \odot (q_{t-x} W_{qy}) + b_y) \\
a_t &= (1 - u_t) \odot q_{t-x} + u_t \odot y_t
\end{aligned}
\tag{9}
$$

where the output of the CNN layer is given as input to this layer, and $x$ is the number of hidden cells. The final layer of the hybrid model is the fully connected layer. The input to the fully connected layer is the hidden states of the skip connection at time $t$, which is represented as $p_t^V$, and $x$ is the hidden states of the skip connection from the timestamp $t - x + 1$ to $t$, which is in the form of $p_{t-x+1}^S, p_{t-x+2}^S \ldots p_t^S$. The final output of the fully connected layer is computed as follows:

$$
p_t^D = W^v p_t^v + Sum \sum_{i=0}^{x-1} W_i^S p_{t-i}^S + b
\tag{10}
$$

where $p_t^D$ is the final prediction result of the hybrid CNN-LSTM model.

The parameter setting of the proposed CNN-LSTM for this experiment are given in Table 4 .

**Table 4.** Different parameter settings.

| Parameters | Value |
| --- | --- |
| Convolutional layer filters | 64 |
| Convolutional kernel size | 1 |
| Convolutional layer activation function | ReLU |
| Convolutional layer padding | Same |
| Pooling layer pool size | 1 |
| Pooling layer padding | Same |
| Pooling layer activation function | ReLU |
| Number of LSTM hidden cells | 128 |
| Number of skip connections | 2 |
| LSTM activation function | tanh |
| Batch size | 32 |
| Loss function | RMSE, MAE |
| Learning rate | 0.0001 |
| Epochs | 100 |

The data are first fed into a one-dimensional convolution layer, which extracts features and produces a three-dimensional output vector (None, 10, 64), with 64 being the size of the convolution layer filters. The vector then enters the pooling layer, where it is converted into a three-dimensional output vector (None, 10, 64). The output vector is then fed into the LSTM layer for training, and the output data (None, 128) from the previous is fed into another complete connection layer after training to acquire the output value; 128 is the number of hidden units in the LSTM layer. Skip connections are added between LSTM cells that receive input from the previous layers. Our hybrid CNN-LSTM has the settings as shown in Table 4. The hyperparameter settings for different experiments are summarized in the Table 5.

**Table 5.** Different parameter settings.

| Hyperparameters | LSTM | CNN | CNN-LSTM |
|---|---|---|---|
| Model Nodes | 3 | 64 | 64 |
| Batch Size | 32 | 32 | 32 |
| Optimizer | ADAM | SGD | ADAM |
| Learning Rate | 0.001 | 0.001 | 0.001 |
| Training Data | 70% | 70% | 70% |
| Validation Data | 10% | 10% | 10% |
| Test Data | 20% | 20% | 20% |

### 5.5. Optimization Approach

We adopt the same optimization approach as that of the traditional time-series forecasting. Consider a time series of the form $X^t = [x_1, x_2, \ldots, x^t]$, with a sizeable window size $g$ at timestamp $t$. The resulting time series is formulated as $Y^t = [x_{t-g+1}, x_{t-g+2}, \ldots, x_t]$. Thus, the problem becomes a classification task with a defined set of feature-value pairs and is solved using [40].

## 6. Experiments and Results

We conducted extensive experiments with three methods including our new method to prove the effectiveness of our proposed hybrid CNN-LSTM method. We compared our method with CNN and LSTM using the same training set and test set under the same experimental setup. All the experiments are performed in an Intel i7-8750H Windows 10 environment with a single GTX 1050 Ti GPU. The framework used in this work is Keras with a Tensorflow back-end. In order to evaluate the prediction accuracy of the hybrid CNN-LSTM model, the mean absolute error (MAE), root mean square error (RMSE), and R-squared accuracy are calculated:

$$\text{MAE} = \frac{1}{n} \sum_{j=1}^{n} |\hat{y}_j - ty_j| \tag{11}$$

$$\text{RMSE} = \sqrt{\frac{1}{n} \sum_{j=1}^{n} (\hat{y}_j - y_j)^2} \tag{12}$$

$$R^2 = 1 - \frac{\left(\sum_{j=1}^{n} (y_j - \hat{y}_j)^2\right)/n}{\left(\sum_{j=1}^{n} (\bar{y}_j - t\hat{y}_j)^2\right)/n} \tag{13}$$

where $\hat{y}_j$ is the predicted value and $y_j$ is the actual value. Smaller values of MAE and RMSE indicate better accuracy. For the R-squared accuracy, $\hat{y}_j$ is the predicted value, $y_j$ is the real value, and $\bar{y}_j$ is the average value.

*Results*

Table 6 summarizes the different evaluation indexes of all the methods used on the test dataset for all three metrics. The processed dataset was used to train CNN, LSTM, and hybrid CNN-LSTM. The obtained results are shown in Figures 8–11. It can been seen that the hybrid CNN-LSTM has the closest-fitting degree of prediction when compared to the other two methods. The evaluation index of each method is calculated and the comparison results can be seen in Table 6 and Figure 9. It can be seen from Table 6 that the hybrid CNN-LSTM model has the lowest MAE and RMSE values, and the R-squared accuracy is almost close to 1. MAE decreases from 32.985 to 30.125 by 9.084% for LSTM as compared to CNN, R-squared accuracy increases by 2.123% with the *p*-value < 0.05, and RMSE decreases from 45.012 to 42.98 by 4.618%, hence LSTM performs better than CNN. The evaluation index of MAE and RMSE for the hybrid CNN-LSTM are the lowest and the R-squared accuracy is highest. The MAE and RMSE values of hybrid CNN-LSTM model improves by 10.863% and 10.477%, and R-squared accuracy improves by 1.6%. This proves that prediction forecasting performance can be improved using the hybrid CNN-LSTM model.

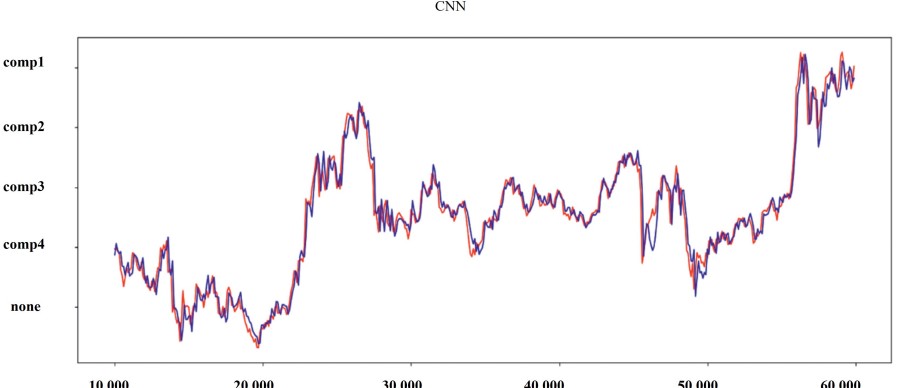

**Figure 8.** Results for the predicted values and real values for CNN.

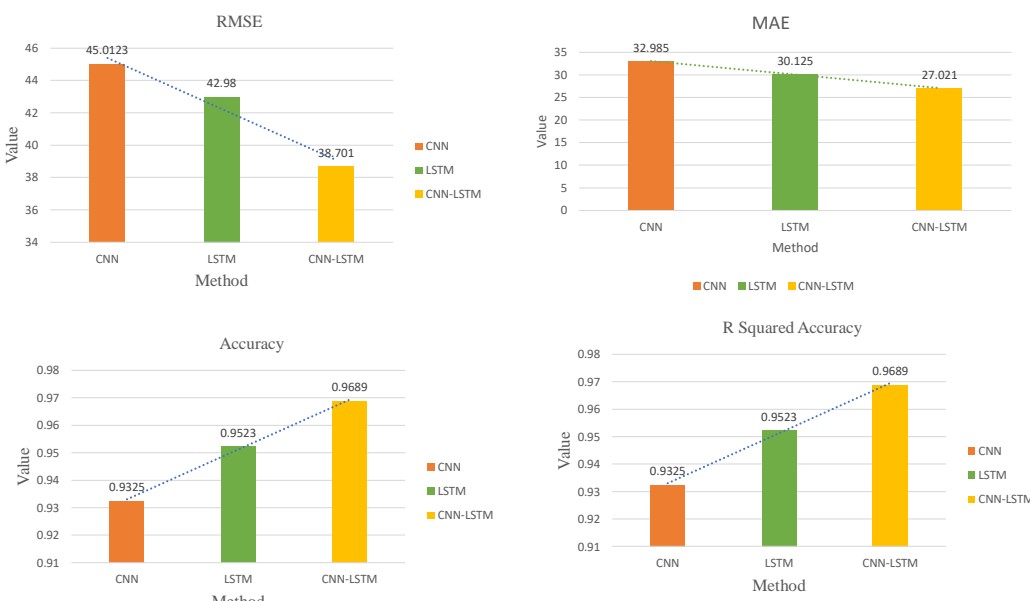

**Figure 9.** Overall comparison of different metrics used to evaluate CNN-LSTM model.

It can be seen from the results that hybrid CNN-LSTM performs best among all the methods. It performs better than the other two methods in terms of error values and fitting degree.

**Table 6.** Comparison of different evaluation indexes for three methods.

| Method | MAE | RMSE | R-Squared Accuracy |
|---|---|---|---|
| CNN | 32.985 | 45.012 | 0.932 |
| LSTM | 30.125 | 42.98 | 0.952 |
| CNN-LSTM | 27.021 | 38.701 | 0.968 |

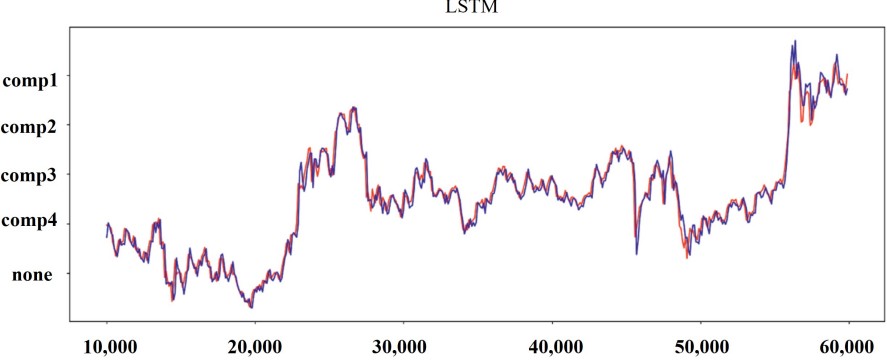

**Figure 10.** Results for predicted and real values for LSTM.

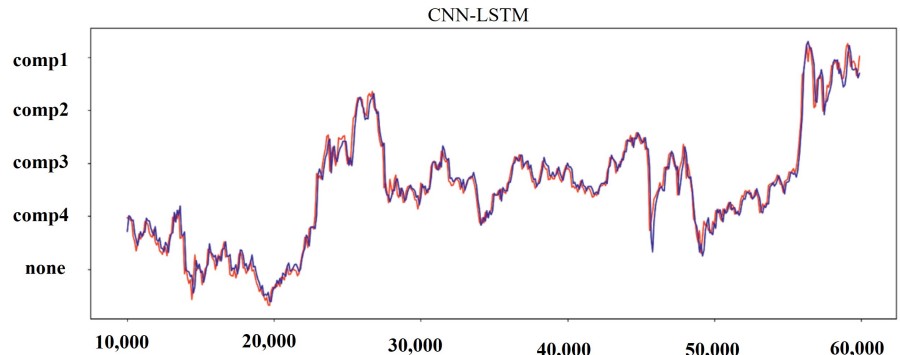

**Figure 11.** Results for predicted and real values for the hybrid CNN-LSTM model.

## 7. Lessons Learned

During the course of this study we came across the following difficulties:

*Data Interoperability*: Currently, data are not shared across machines in a production line. Industry 4.0 is now working on system integration to make data more inter-operable so that better decisions may be made to optimize the production process. Where elements of the knowledge wrapped in the data are made available, we believe data integration can improve machine learning results.

*Match and Validate Data:* It can take a long time to come up with rules to match data received from multiple sources. This becomes progressively difficult as the number of sources grows. Machine learning models can be taught to learn the rules and forecast fresh data matches. There is no limit to the amount of data that may be used, and more data are actually beneficial in fine-tuning the model.

*Modelling:* When compared to using a single ML model, combining multiple ML models can yield superior predictions. However, developing the hybrid objective function for optimization is difficult given the complex nature of time-series data such as missing entries and dynamic periodicity. For example, classification and anomaly detection techniques can be coupled to keep classification model precision while retaining anomaly detection benefits. PdM can be used on equipment or systems that do not have a huge dataset in this fashion.

## 8. Discussion

Machine learning methods are a potential way to proceed if the time series is anticipated to be nonlinear to a substantial extent. Hybrid approaches have the capacity to handle non-linearity in a time series while also embracing the strengths of statistical methods. In a situation where there is non-linearity but also a clear pattern and seasonality, hybrid approaches are a viable choice to examine. If the time series can be assumed to be generally linear, classical statistical methods will be used to their full potential in a simple yet incredibly effective and efficient manner. CNN with LSTM consume more time and are computationally a bit more expensive. The proposed hybrid approach suffers from short-time sequence and real-time predictions. Hybrid models are good choices for the type of data, which has good non-linearity, clear patterns, and seasonality. It was also observed that hybrid models tend to suffer in terms of multi-dimensional data in a smooth fashion. Therefore we will consider further research on multi-dimensional data in terms of data relevance and density.

## 9. Conclusions and Future Direction

In this work we proposed a predictive maintenance system (PdM) based on a hybrid CNN-LSTM model for multivariate time-series classification. The proposed approach uses different features such as voltage, pressure, vibration, rotation, machine age, error type, number of components, model type, and failure as inputs within a full time-sequence of the data. CNN extracts the features from the input data, and LSTM learns from those extracted features. Together the hybrid model is able to capture the long-term and short-term data patterns for robust predictions. We used Microsoft data to verify our experiments. Results show that the hybrid CNN-LSTM has the best prediction accuracy and performs better compared to CNN and LSTM. In addition, the evaluation indexes MAE and RMSE are the smallest of all methods, and R-squared accuracy is close to 1. The hybrid model, if implemented, can maximize the production process in any given smart manufacturing production line. This work can be further extended in a number of ways. The performance of model can be checked with different data characteristics. The model could be implemented for probabilistic PdM and compared with some existing methods. The integration of feature information and feature dimensions into LSTM with a skip connection is still a challenge.

**Author Contributions:** Conceptualization, A.W., M.A.I. and J.G.B.; methodology, A.W.; validation, A.W. and M.A.I.; formal analysis, A.W.; investigation, A.W.; writing—original draft preparation, A.W.; supervision, M.A.I. and J.G.B.; project administration, M.A.I.; funding acquisition, M.A.I. and J.G.B. All authors have read and agreed to the published version of the manuscript.

**Funding:** This publication has emanated from research supported in part by a research grant from Science Foundation Ireland (SFI) under Grant Number SFI/16/RC/3918 (Confirm), and also by a research grant from Science Foundation Ireland (SFI) under Grant Number SFI/12/RC/2289_P2 (Insight), with both grants co-funded by the European Regional Development Fund.

**Institutional Review Board Statement:** Not applicable.

**Informed Consent Statement:** Not applicable.

**Data Availability Statement:** The dataset is open-sourced to the researchers from the Microsoft at https://github.com/ashishpatel26/Predictive_Maintenance_using_Machine-Learning_Microsoft_Casestudy/tree/master/data.

**Conflicts of Interest:** The authors declare no conflict of interest.

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
