# Peer review of "Prediction of Machine Failure in Industry 4.0: A Hybrid CNN-LSTM Framework"

_applsci, doi:10.3390/app12094221_

Round 1
Reviewer 1 Report
The authors have presented a hybrid CNN-LSTM framework for predictive maintenance of machines. Its an interesting work. The following comments need to be addressed for improving the manuscript.
1) The sensor data need to be presented visually. Plot relevant sensors data for better understanding.
2) Need to describe different error codes/ fault conditions considered in the experiments.
3) Tabulate the details of training and test data set (no of samples).
4) Tabulate the input parameters and output parameters used for prediction.
5) Did the authors experimented LSTM alone with skip connection approach for checking the effectiveness with CNN-LSTM with skip connection.
6) Tabulate the hyper parameters used in all the experiments.
7) Include conclusion section at the end of the manuscript.
Author Response
Thank you reviewing our manuscript, with an opportunity to improve. The manuscript quality has been improved by addressing your valuable and positive comments. Please find the attached file in response to all the aforementioned comments and suggestions.

Reviewer 2 Report
This work proposes a deep learning based method for multivariate time-series forecasting for Predictive Maintenance (PdM).
Comments:
- In the abstract, add 1-2 sentences on the main numerical findings.
- In the Introduction section, the contributions are clear but what about novelty? Is the proposed hybrid architecture novel? Several articles used hybrid CNN/LSTM networks before.
- Expand the discussion on related works. Discuss „Vibration-based anomaly detection using LSTM/SVM approaches“, „Bearing fault diagnosis using transfer learning and optimized deep belief network“, „Acoustic anomaly detection of mechanical failures in noisy real-life factory environments“. Summarize the limitations of existing methods as a motivation for your study.
- Section 3: support the relevance of the scenario by references.
- Explain the data processing step in more detail. How do you deal with dataset imbalance?
- Line 188: a reference is missing.
- Explain all the training settings and parameters such as learning rate, etc. Did you use cross-validation? What was the training/testing ratio split?
- For R-squared values also present the p-values.
- Compare your results with the results of other authors achieved on the same dataset.
- Add the discussion section and discuss the limitations of the proposed methodology.
Author Response

(The authors gave the same response as above.)

Round 2
Reviewer 2 Report
The authors have addressed all my comments and revised the manuscript accordingly. The paper can be accepted for publication.